# Development and Applications of Hydrogel-Based Triboelectric Nanogenerators: A Mini-Review

**DOI:** 10.3390/polym14071452

**Published:** 2022-04-02

**Authors:** Sheng-Ji Wang, Xin Jing, Hao-Yang Mi, Zhuo Chen, Jian Zou, Zi-Hao Liu, Pei-Yong Feng, Yuejun Liu, Zhi Zhang, Yinghui Shang

**Affiliations:** 1Key Laboratory of Advanced Packaging Materials and Technology of Hunan Province, Hunan University of Technology, Zhuzhou 412007, China; wangsj418@163.com (S.-J.W.); cz100520@163.com (Z.C.); lzh_132000@163.com (Z.-H.L.); fpyedu@163.com (P.-Y.F.); yjliu_2005@126.com (Y.L.); 2National Engineering Research Center for Advanced Polymer Processing Technology, Zhengzhou University, Zhengzhou 450000, China; zj980428@163.com; 3Shenzhen Weijian Wuyou Technology Co., Ltd., Shenzhen 518102, China; zhangzhiuse@163.com (Z.Z.); shyh1103@163.com (Y.S.)

**Keywords:** hydrogel-based TENGs (H-TENGs), electrodes, flexible sensors, energy harvesting, biomedical applications

## Abstract

In recent years, with the appearance of the triboelectric nanogenerator (TENG), there has been a wave of research on small energy harvesting devices and self-powered wearable electronics. Hydrogels—as conductive materials with excellent tensile properties—have been widely focused on by researchers, which encouraged the development of the hydrogel-based TENGs (H-TENGs) that use the hydrogel as an electrode. Due to the great feasibility of adjusting the conductivity and mechanical property as well as the microstructure of the hydrogels, many H-TENGs with excellent performance have emerged, some of which are capable of excellent outputting ability with an output voltage of 992 V, and self-healing performance which can spontaneously heal within 1 min without any external stimuli. Although there are numerous studies on H-TENGs with excellent performance, a comprehensive review paper that systematically correlates hydrogels’ properties to TENGs is still absent. Therefore, in this review, we aim to provide a panoramic overview of the working principle as well as the preparation strategies that significantly affect the properties of H-TENGs. We review hydrogel classification categories such as their network composition and their potential applications on sensing and energy harvesting, and in biomedical fields. Moreover, the challenges faced by the H-TENGs are also discussed, and relative future development of the H-TENGs are also provided to address them. The booming growth of H-TENGs not only broadens the applications of hydrogels into new areas, but also provides a novel alternative for the sustainable power sources.

## 1. Introduction

The triboelectric nanogenerator (TENG), first reported by Wang et al. in 2012 [1], is an invention based on the coupling between electrostatic induction and frictional electrical effects. The TENG can essentially be regarded as a kind of capacitive variable electric-field source, and its output power is related to the triboelectric charge density [2,3]. TENGs can harvest mechanical energy that is ubiquitous in human daily life, including wave [4], wind [5], as well as human motion such as respiration and body movements [6]. TENGS have unique superiority in simple design, low costs, feasible portability over other power sources [7,8,9,10,11]. The remarkable advantages of TENGs endow them with great potential in artificial intelligence [12,13], wearable electronic devices [14], internet of things [15], and biomedical devices [16,17]. Excellent achievements have been realised in the research of TENGs in terms of the rapid demand for green and sustainable energy in portable wearable devices.

Conventional TENGs always employed rigid materials as electrode materials. Those kind of TENGs were not able to withstand large strains due to the high rigidity of the electrode material [18,19,20], which greatly limited the applications of TENGs, especially in the portable devices. To address this concern, many alternative flexible materials were developed to fabricate the flexible TENGs, such as polymer nanofibers [21], commercial fabrics [22], papers [23], foams [24], etc. Compared to other alternatives, hydrogels are soft hydrophilic polymeric materials with three-dimensional (3D) cross-linked networks [25,26]. Hydrogels are ideal materials for flexible conductors due to their good biocompatibility, tunable physical/chemical properties, and excellent electrical conductivity with the addition of conductive fillers (e.g., metal particles, conductive polymers, carbon materials, salts) and stimulation responsiveness [27,28,29,30,31,32,33]. Therefore, hydrogels have received great attention in terms of developing flexible and portable TENGs in recent years.

The working principles of traditional TENGs are as follows: when the tribopositive layer contacts the tribonegative layer, contact electrification occurs between the two tribolayers, generating an equal number of opposite charges on the surface. Once separated, electrostatic potential difference was created on the surface of the two tribolayers. Meanhile, the design of H-TENGs was slightly different to that of conventional ones, in which the hydrogel that acted as conductor was usually encapsulated in elastomeric materials, and then connected to wires as electrodes for TENGs [34,35,36,37,38,39]. According to the different transportation in conductivity of conductive components, hydrogel conductors can be further divided into ionic and electronic ones, which result in different charge transfer behaviors as demonstrated in Figure 1 [40,41,42]. For the ionic ones, the negative ions cause the redistribution of the ions in the hydrogel; positive ions move toward the negative charges, while the negative ions move toward the opposite direction. Negative ions accumulate around the wire, causing the wire to polarize and the same number of positive charges to appear, while negative charges flow through the wire in the opposite direction and achieve output performance. The flow of positive and negative ions is reversed when the two tribolayers contact again, and the output performance is generated simultaneously. Meanwhile, for electronic H-TENGs, the working principle is similar to that of traditional TENGs, relying on the transfer of electrons to generate the output performance.

Although some reviews have emerged on the development and working principles of TENGs, the H-TENGs still have a long way to go in terms of practical applications. In this review, we highlight materials research that is highly related to the performance of the H-TENGs, including the composition of the hydrogel networks, the engineering between the hydrogel and its sealants, as well as the retention of water in hydrogels. Moreover, we also overview the H-TENG devices that have been reported in the literature to date. Furthermore, we also highlight the challenges and opportunities in the field of H-TENGs.

## 2. Preparation Strategy of H-Electrodes

Based on the network structure of hydrogels, the H-electrodes used in H-TENGs are classified into single-network (SN), multi-network (MN), and nanocomposite (NC) hydrogels, according to their composition. Table 1 lists an overview of the H-TENG studies.

### 2.1. SN H-Electrodes

The SN hydrogel is prepared with a single kind of polymer—which can obtain good conductivity when loaded with inorganic salts such as sodium chloride (NaCl) [66] and lithium chloride (LiCl) [67]—being swollen in water. These hydrogels demonstrate significant advantages concerning the feasibility of preparation and good flexibility, suggesting their wide usage as electrodes for TENGs.

Xu et al. [43] (Figure 2a) prepared a polyvinyl alcohol (PVA) hydrogel by the freeze-thaw method and encapsulated it with a Polydimethylsiloxane (PDMS) film bag with nickel fabric to form a hemispherical structure to be used as an electrode for a TENG. This TENG (8 cm × 8 cm) achieved a peak output voltage and current of 200 V and 22.5 µA, and peak output power of 2 mW at a load resistance of 10 MΩ, respectively. In addition, the tube-shaped H-TENG was designed and used to harvest energy from human motions. Moreover, Parida et al. [44] prepared a highly transparent PVA hydrogel with borax as a crosslinker, which was then integrated with silicone rubber and VHB tape to prepare the H-TENGs (Figure 2b). Besides their high stretchability and good transparency, the H-electrodes of this TENG also demonstrated great self-healing efficiency attributed to the dynamic hydrogen bonding and borate ester linkages in the borax cross-linked PVA hydrogel, which enabled the TENG to self-heal and to obtain an effective and stable output signal even after self-healing. Meanwhile, the TENG also displayed long-term durability for 1000 continuous cycles of voltage output measured at 15-day intervals. To enhance the long-term stability of the H-TENGs, Pu et al. [40] introduced LiCl into the Polyacrylamide (PAAm) hydrogel owing to its great water retention ability, which also played a conductive role in the resultant H-TENGs after being encapsulated with PDMS layer. This TENG demonstrated 1160% stretchability and 96.2% transparency. In addition, the TENG was able to achieve an instantaneous peak power density output of 35 mW/m^2^ with a stable output performance even after 5000 cycles. The stability was also able to be preserved between 0 to 80 °C.

Conventional hydrogels are prone to dehydration at room temperature or high temperature. Therefore, when being used as an electrode of a TENG, they always need to be encapsulated with elastomer, which were also called the triboelectric materials. However, due to the difference in physicochemical properties, the bonding between the hydrogel and the encapsulation layer is always weak and prone to peeling off during usage, leading the hydrogel to lose water, thus further shortening the service life of the assembled TENGs. Therefore, to address this problem, our group [45] (Figure 2c) used benzophenone (BP) grafting as a bridge to enhance the interfacial bonding between hydrophobic triboelectric elastomer material and hydrophilic hydrogel. The adherence between the hydrogel and elastomer was greatly improved. This allows the prepared TENG to operate steadily at high frequencies of 20 Hz and the generated energy to be successfully stored and used to power small electronics, such as timers, pedometers, and digital watches. Furthermore, the developed TENG maintained approximately 56% transparency and provided a high output voltage and output current of 311.5 V and 32.4 μA, respectively. Moreover, a maximum power density of 2.7 W/m^2^ was also achieved at an external load of 4.7 MΩ. To further reduce the water effect on the stability of the hydrogel, Lv et al. [46] (Figure 2d) used ionic liquids, 1-butyl-2,3-dimethylimidazolium bis(trifluoromethylsulfonyl)imide ([BMMIm][NTf2]), as solvents to prepare the hydrogel, and encapsulated it with PDMS to assemble the TENG. Due to the ultra-long stability under the different temperature and humidity possessed by the ionic liquid, the TENG maintained stable electrical signal output in the temperature and humidity range of −25 to 60 °C and of 20% to 80%, respectively. Furthermore, the developed TENG based on the ionic liquid hydrogel displayed stable electrical output after 90 days of storage.

Enlarging the contacting area is also a feasible way to increase the output power of the H-TENGs. For example, Qi et al. [47] (Figure 3a) used hierarchically wrinkled electrospun polycaprolactam (PA6) membranes to decorate the PAAm/LiCl hydrogel, and obtained efficient and stretchable TENGs for mechanical energy harvesting. In this design, the PA6 nanofiber membrane was collected on the hydrogel by being attached on the balloon as a collector, which not only achieved high stretchability, but also expanded the triboelectric area and improved the output performance of the device due to the rough surface formed by the nano- and micro-structure. In this assembled TENG, the electrical charge was generated by contact electrification between the PA6 membrane and the Ecoflex substrate on the inner cavity surface under external mechanical force, and the PAAm hydrogel in this structure acted as an electrostatic balancing material. The output voltage and current values upon compressive forces achieved 270 V and 11 μA, respectively. Moreover, Li et al. [48] (Figure 3b) investigated the effect of the contact area between the carbon tape and the hydrogel on TENG output performance. They chose a single network hydrogel composed of PVA and phosphoric acid as an electrode in the H-TENGs. Here, the hydrogel can be regarded as an external capacitor which highly affects the performance of TENGs. According to $C=\varepsilon_{0}\varepsilon_{r}A/d$, a higher output capacity would be achieved when the contact area was larger. Thus, it was reported that the output capacity gradually increased with the increase of the contact area between carbon tape and hydrogel, and that the highest output voltage, output current, and power density was achieved to 992 V, 44.8 μA, and 26 W/m^2^, respectively, at last. Furthermore, the high output was still maintained even after six months, with this having been contributed to by the superior water retention of the H-electrode, which showed great potential in stretchable energy harvesters and self-powered stretchable sensors.

Although there have been lots of H-TENGs using SN hydrogels as H-electrodes, and despite them having demonstrated promising performances as sustainable power sources, due to their limited composition and adjustability, the SN hydrogels were further improved by introducing additional networks via physical or chemical treatments, and were named as MN hydrogels.

### 2.2. MN H-Electrodes

In contrast to SN hydrogels with a single kind of polymer, MN hydrogels were endowed with multi-functionality via the introduction of multiple polymers containing different functional groups. Compared to the SN hydrogels, although the preparation of MN hydrogels was more complicated, the properties of MN hydrogels were more diversified, and the assembled TENGs also demonstrated great advantages. For example, loading conductive polymer poly(3,4-ethylenedioxythiophene):poly(styrene sulfonate) (PEDOT:PSS) into the hydrogels not only brought good electrical conductivity to the hydrogel without salt, but also improved the mechanical properties through entanglement and hydrogen bonding [68,69,70,71]. Besides that, polydopamine (PDA), polyvinyl alcohol (PVA), polyacrylic amide (PAAm), and polyacrylic acid (PAA) were also combined to assemble the hydrogel with self-healing ability and adhesiveness [72,73,74,75]. Cellulose derivatives, such as cellulose nanofibers and hydroxyethyl cellulose, were also commonly employed into the hydrogel systems to prepare the MN hydrogels. Lots of hydrogen bond acceptors emerged in large numbers, which, along with cellulose, can form many hydrogen bonds with free and bonded water molecules in the hydrogel, which is helpful to enhancing the anti-freezing ability of the hydrogel [52].

To prepare the hydrogel with anti-freezing ability, Bao et al. [52] (Figure 4a) introduced hydroxyethyl cellulose into the hydrogel system, which was synthesized by one-step radical polymerization of acrylamide monomer in aqueous hydroxyethyl cellulose solution, which can tolerate −69 °C without freezing after the addition of lithium chloride. It was found that the hydroxyethyl cellulose as a physical cross-linking agent not only enhanced the mechanical properties, but also provided water retention properties. The anti-freeze H-TENG had a constant elongation of 150%, and successfully harvested human biomechanical energy to drive wearable electronics, even in harsh environments, with an instantaneous peak power density of 626 mW/m^2^ on an external resistor when the frequency was 2.5 Hz. The H-TENG with an area of 3 × 3 cm^2^ also achieved an output open-circuit voltage, short-circuit current, and short-circuit transfer charge of 285 V, 15.5 µA, and 90 nC, respectively. This work is expected to provide a promising approach for the development of flexible energy sources under harsh conditions.

The introduced conductive polymers not only endow the MN with good conductivity but also better mechanical performance. Zhang et al. [76] introduced PEDOT:PSS to γ-polyglutamic acid (PGA) hydrogel, which demonstrated a significant increase in mechanical strength from 43 to 380 kPa, in conductivity from 1.3 to 12.5 S/m and in elongation from 283% to 650%, respectively (Figure 4b). Moreover, the conductive hydrogel encapsulated by PDMS as top layer was directly attached onto human skin to form H-TENG, and was demonstrated to be able to communicate Morse code by tapping with fingers. Similarly, Sun et al. [31] (Figure 4c) introduced PEDOT:PSS into the PAM hydrogel, and successfully prepared a tough hydrogel with 298 kPa in tensile strength and 2850% in elongation. They also used the prepared hydrogel as an electrode to assemble the H-TENG, which was sandwiched by PU tape. This H-TENG exhibited high output performance with open-circuit voltage, short-circuit current, and short-circuit transfer charge reaching 383.8 V, 26.9 µA, and 92 nC, respectively. In addition, the H-TENG had achieved stable output performance at 0% to 300% tensile strain and maintained stable output in 16,000 compression-release cycles. These excellent performances also demonstrated the feasibility of the H-TENG prepared by this method for practical applications.

To enhance the service life of the TENGs, self-healable hydrogels are today used as electrodes to prepare H-TENGs. For example, Long et al. [53] introduced the self-polymerized dopamine into the PAM hydrogel system to prepare PDA/PAM hydrogels with self-healing ability. The dual network of PAM and PDA endow the hydrogels more than 60 times stretching and good self-healing ability. The healed hydrogel almost preserved its output properties, which only demonstrated a 10% decrease in the current and voltage and a 20% decrease in the charge density, implying that the developed H-TENGs display great self-healing ability. In addition, by sandwiching the hydrogel with the acrylic substrate and PTFE or VHB as triboelectric layer, this H-TENG can provided enough power for commercial electronic devices such as watches or LED lights when it works. Likewise, Liu et al. [77] (Figure 5a) further enhanced the output performance of the H-TENGs by integrating the self-healable hydrogel electrodes with a modified triboelectric layer—in which the hydrogel was prepared by using PVA, PDA, chitosan, and agarose—which has a superb self-healing ability that can heal within 1 min. The healing efficiency reached 98%, and the triboelectric layer was modified by loading CNT into PDMS via chemical vapor deposition. It was proved that the self-healable hydrogel endows the healing ability of the H-TENGs, whose output performance could reach 94% after healing. Moreover, the composite triboelectric layer composed of PDMS and CNT also made a great contribution to the output performance of the H-TENGs, which displayed as two times higher than that of H-TENGs with pure PDMS layer, owing to the higher dielectric constant and lower dielectric loss of the PDMS/CNT compared to PDMS.

Moreover, it was also possible to adjust the output performance of the H-TENGs by adjusting the mechanical behavior of the hydrogels. For example, our group [27] (Figure 5c) fabricated a series of PVA/SA DN hydrogels by adjusting the concentration of the sodium alginate (SA) in the hydrogel, which successfully controlled the viscoelastic properties of the hydrogel system. The H-TENGs were subsequently fabricated with over 90% transparency and 250% stretchability after encapsulation by PDMS bags, respectively. The prepared TENGs were able to achieve peak output voltage and current of 203.4 V and 17.6 µA, with a power density of 0.98 W/m^2^ at an external resistance of 4.7 MΩ. The TENG can also achieve more than 85% of the original output performance after storing for 28 days. Additionally, it was confirmed that the higher elastic hydrogel is more favorable for electrodes in TENGs. Moreover, similar phenomena were also reported in the hydrogels with metal coordination bonds. For example, Sheng et al. [54] prepared the metal coordinated SA/PAM-co-PAA MN hydrogel, in which Zn ions were introduced to form dynamic coordination bonds with carboxyl groups of SA and PAA, which endowed the H-TENGs great stretchability and stable outputting performance of 800%, as well as stable signal output over 16,000 cycles (Figure 5b).

### 2.3. NC H-Electrodes

There has been wide study of the preparation of NC hydrogels which were fabricated via the incorporation of nanoparticles into hydrogels. The intervention of various nanoparticles—such as conductive nanofillers such as graphene [78], carbon nanotube [79], metallic nanoparticles [80], and Mxene [81]—could endow the hydrogel with different properties, making it demonstrate good conductivity but also higher mechanical performance. Some nanoparticles even promote the polymerization of hydrogels [59,82,83,84,85].

Li et al. [57] added zinc oxide (ZnO) nanoparticles as nano-fillers into the PAA ionic gel to prepare the PAA/ZnO NC hydrogel, and formed entanglements between ZnO nanoparticles and PAA molecular chains that enhanced the mechanical properties of the hydrogel (Figure 6a). The H-TENG prepared using this NC hydrogel as electrode demonstrated an ultra-wide temperature operating range from −30 °C to 80 °C and a long-term stability of more than 30 days. Moreover, the resultant H-TENG maintained a high output performance of more than 200 V after 10,000 cycles of testing. In addition, the zinc ion in zinc acetate dihydrate forms a reversible metal coordination bond with the carboxyl groups of PAA, which enables the ionic gel electrode to maintain its original output capability after repeated cutting/healing experiments. Owing to its excellent mechanical properties, stability, and self-healing properties, the assembled H-TENG prepared using this ionic gel was highly reliable and had a long service life.

Moreover, Xu et al. [58] used nanoclay as nanofillers to prepare the PAM NC hydrogel. In the system, nanoclay not only enhanced the mechanical property of the PAM hydrogel, but also accelerated the crosslinking process of the hydrogel. Next, to eliminate the side effect of water onto the long-term performance of the H-TENGs, the prepared hydrogel was subsequently soaked with a binary solvent of ethylene glycol/water to obtain the organohydrogel, and then encapsulated using an elastomer to assemble the H-TENGs. Numbers of hydrogen bonds formed between the introduced nanoclay and PAM enable the hydrogel to self-heal after mechanical damage and maintain its original output performance. At the same time, the anti-freezing ability of the organohydrogel endowed by the absorbed glycol allowed the H-TENG to be operated normally at −30 °C, and maintained a stable output performance after 30 days. Furthermore, to prepare the whole self-healable H-TENGs, besides the preparation of self-healable electrodes, triboelectric layers with self-healing efficiency have also emerged. Huang et al. [59] synthesized healable poly(dimethylsiloxane) (IU-PDMS) elastomers as the encapsulation layer of the hydrogel by cross-links of imine bonds and quadruple hydrogen bonds (Figure 6b), which were formed by grafting 2(6-isocyanatohexylaminocarbonylamino)-6-methyl-4[1H]pyrimidinone to PDMS at the bis(3-aminopropyl) capped end and then reacting with 1,1,1-tris[(4-formylphenoxy)methyl]ethane. Owing to the great self-healing ability in the organohydrogel as reported previously and IU-PDMS, the assembled H-TENG demonstrated excellent self-healing performance, and which self-healing efficiency was almost 100% in output voltage and current. In addition, the prepared H-TENGs also displayed working ability in harsh conditions such as extremely cold or hot environments, which showed great potential for practical applications.

Besides the reinforcing effect of nanofillers mentioned above, conductive nanofillers could kill two birds with one stone; they could not only enhance the mechanical properties of the NC hydrogel, but could also improve its conductivity. For example, Luo et al. [60] introduced MXene into the borax cross-linked PVA hydrogels as an electrode to assemble H-TENG, which not only enhanced the hydrogel’s elongation by 1.4 times, but also improved the H-TENG’s open-circuit voltage by 4 times. The TENG which was prepared on the basis of the hydrogel obtained an output voltage of 230 V under a single electrode mode of operation, and had good stretchability and self-healing property (Figure 7a). In addition, to further enhance the conductivity of the hydrogel, Wang et al. [30] introduced silver nanowires (AgNW) into ionic chitosan hydrogels to prepare composite hydrogels (Figure 7b). The chitosan/AgNW hydrogel cross-linked by copper ions was used to assemble the H-TENG, which demonstrated the output voltage and current density of 218 V and 34.44 mA/m^2^, respectively. Meanwhile, the H-TENG was able to reach 88% of the original output voltage after one week of storage, and good performance was maintained even under harsh environments.

The incorporation of these nanoparticles has also been widely used in the development of high-performance hydrogels, as they provide properties for hydrogels far beyond those achieved by simple polymer network design [86], which has facilitated the use of nanocomposite hydrogels for the preparation of high performance TENGs.

## 3. Applications

Hydrogel-based TENGs have attracted a wave of research due to their excellent flexibility, tunable mechanical strength, and ability to achieve self-healing. Hydrogel-based TENGs hold great promise in many fields, such as flexible sensing, bioenergy harvesting, and biomedical applications [87,88,89]. In this section, we will present some representative applications of the TENGs based on very recent studies.

### 3.1. Self-Powered Sensors

Flexible sensors are made of materials that can be extended to a certain extent without changing their properties [90]. TENGs assembled based on H-electrodes not only retain the good flexibility of hydrogels, but also provide good electrical conductivity for TENGs. In the current research, H-TENGs have been widely used in the fields of motion monitoring, haptic perception, and human-computer interaction due to their combination of sensitivity, stretchability, and conformability.

Sheng et al. [54] prepared an MN H-TENG sensor which was able to recognize the motions of different parts of the human body, such as the wrists and fingers, and which demonstrated good linearity under external tensile stress from 0 N to 50 N, which also demonstrated the reliability of this self-powered sensor (Figure 8a). In addition, a real-time demonstration of the process in a practical application of a self-powered smart elastic band system, including a signal acquisition and processing system, and a real-time software output interface, demonstrate the potential of this TENG for human motion monitoring and energy harvesting applications.

Luo et al. [60] prepared an NC hydrogel and used this as a sensor encapsulated with elastomer. The self-powered sensor generated different output voltage signals depending on the degree of strain, and can generate corresponding output voltage signals on the different parts of the human body based on different amplitude changes. In addition, the H-TENG being a self-powered tactile sensor, it could recognize different words written on the surface of the TENG. This means that it possesses great potential for applications in wearable self-powered devices and high precision written stroke recognition applications.

Zhao et al. [91] prepared a dual-electrode TENG by sandwiching PDMS with microstructure patterns in the middle with the prepared dual-network ionic gel (Figure 8b). The TENG was able to recognize the pressure change even under different stretching conditions. In addition, it also displayed good reliability in monitoring human motions, and could even monitor breathing and pulse beating.

### 3.2. Energy Harvesting

H-TENGs have excellent flexibility, electrical conductivity, and self-healing properties, and can easily derive energy from deformation. Therefore, H-TENGs are widely used for making wearable electronic devices to harvest biomechanical energy.

Liang et al. [51] prepared a H-TENG with mechanoluminescence property by adding Cu-doped ZnS particles into Ecoflex (Figure 9a). This special design of the TENG was able to kill three birds with one stone: firstly, function as smart skin displayed superb long stability; secondly, capability of detecting pressure limit of 0.58 kPa with a sensitivity of 0.23 kPa^−1^; and thirdly, visualization of the magnitude and location of force via light-emitting response. Meanwhile, this H-TENG could harvest the mechanical energy from the human body and provide energy for commercial electronics. This work exhibited tremendous promise for developing self-powered epidermal electronics with great conformability and multifunctionality. Moreover, Shuai et al. [92] prepared PNA hydrogel by photoinitiation using N-acryloylglycinamide and acrylamide as the hydrogel network. The hydrogel was transformed from a gel-state to a viscous sol-state at 80 °C because of the breaking of the hydrogen bonds. On the contrary, it solidified from the sol-gel state to the gel state after cooling due to the re-formation of hydrogen bonds. The sol-state PNA hydrogel precursor was subsequently extruded into a room temperature ethyl acetate solution through a special mold to cure into PNA gel fibers, which were subsequently coated with poly(methacrylate) and woven into a fabric for energy harvesting, and it was successfully used to power electronic watches.

Moreover, besides biomechanical energy, there are other available sources for energy harvesters, such as water. Qian et al. [34] collected energy from waves and further proposed a feasible future way of developing H-TENGs as sustainable and green energy collectors (Figure 9b). They proposed octopus tentacle-inspired micropatterns on the hydrogel and the triboelectrification layer. It was found that the micropattern on the Ecoflex as triboelectric layer was favorable for the enhancement in the outputting capability of the H-TENGs. When the prepared honey-shaped micro patterned polyampholyte (PA) hydrogel was combined with the H-TENGs, this could act as an adhesive layer on the smooth glass substrate, providing a stable output performance as water slapped against the assembled H-TENGs.

### 3.3. Biomedical Applications

H-TENGs are often used for biomedical applications due to their biocompatibility and ease of preparation into wearable devices, such as drug delivery to stimulate wound healing or accelerating wound healing through electrostatic stimulation. Therefore, H-TENGs for biomedical applications are currently being extensively investigated.

Jeong et al. [88] encapsulated the organohydrogel in silicone to form an elastic microtubular structure that is both a stretchable wire and a type of wearable generator after being braided (Figure 10a). The elastomeric film encapsulated with hydrogel was prepared as a wound patch, which both a wound dressing and an H-electrode. The generation of voltage during movement leads to the generation of an endogenous electric field at the patch, which can electrically stimulate the wound to promote wound healing, significantly increasing the healing rate relative to a wound coated with ordinary hydrogel.

Wu et al. [93] designed a soft hydrogel patch with side-by-side electrodes for non-invasive iontophoresis therapy (Figure 10b). Proof-of-concept experiments with dyes as model drugs on pig skin successfully demonstrated the feasibility of the proposed system. This work not only extended the application of TENGs in the biomedical field, but also provided a cost-effective solution for non-invasive, electrically assisted iontophoretic drug delivery with closed-loop sensing and therapeutic functions.

## 4. Prospect and Conclusions

Great progress has been made in the development of TENGs in the past few years. Here, we present a short review of the recent advances in the H-TENGs, mainly focusing on the preparation strategies of H-electrode materials and their applications for H-TENGs. The excellent mechanical properties, self-healing ability, and output performance of the hydrogels make assembled TENGs fully applicable to the production of foldable, stretchable, and flexible electronic devices, as well as self-powered wearable electronic devices, which can be attached to various parts of the human body or hidden in accompanying clothing for energy harvesting in various scenarios.

In addition, related systems following the H-TENGs have been also developed, enabling remote sensing and human-machine interactions. Moreover, notable progress of the H-TENGs has also been achieved in biomedical applications. To address the problem of the water effect on the stability of the H-TENGs, special designs and organohydrogel have been developed to enable the H-TENGs to be used for long periods. However, we are still a long way from developing novel H-TENGs to address current challenges. First, compared to the traditional electric power devices, recently reported H-TENGs displayed lower power density, which could be further achieved by increasing their charge density by decreasing the dissipation of charges, as well as enhancing the charge-trapping and charge-transportation ability of the conductive hydrogel by introducing the novel nanofillers. Second, before their practical application, the long-term stability of the H-TENGs should be explored over months or even years, rather than over only a few hours in the laboratory. Finally, microstructured hydrogels are favorable for enhancing the outputting capacity of the H-TENGs via enhancing the contacting area. Moreover, it was also desirable to develop a feasible strategy to fabricate the H-TENGs in a scaled way. Therefore, it is highly worth addressing the working principles of H-TENGs, as well as developing their material design. This requires the joint efforts of researchers from different fields. It is believed that the continued progress of H-TENGs will open the door to the reality of flexible wearable electronic devices, and will also provide new ideas for green and sustainable energy harvesting devices.

## Figures and Tables

**Figure 1 polymers-14-01452-f001:**
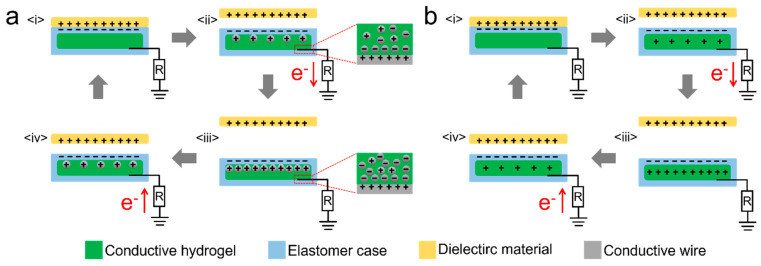
Schematic of working mechanism of the (**a**) ion-conductive hydrogels and (**b**) electron-conductive H-TENG in single-electrode mode.

**Figure 2 polymers-14-01452-f002:**
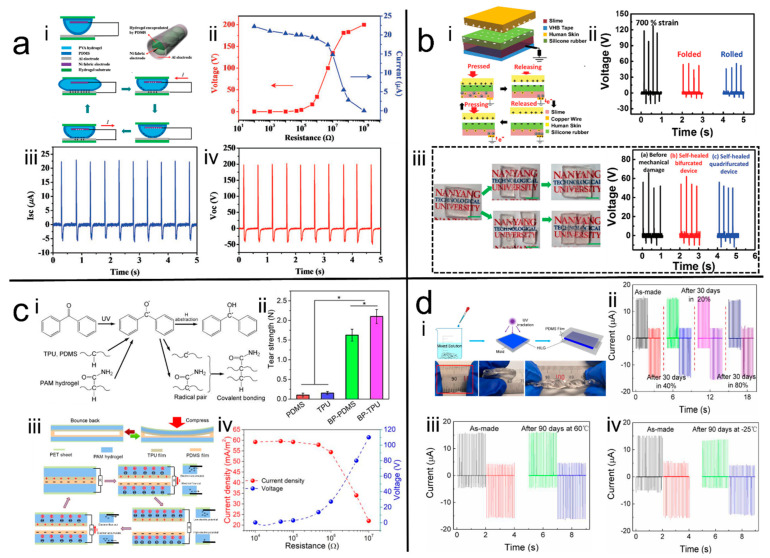
(**a**) <i> Schematic of the hemispherical and tube-shape hydrogel-based TENG, <ii>, <iii> and <iv> performance of the prepared H-TENG [43]. Copyright 2017, Wiley Online Library. (**b**) <i> Schematic of stretchable, transparent and self-healing hydrogel-based TENG, <ii> demonstration of H-TENG flexibility, <iii> self-healing properties of H-TENG [44]. Copyright 2017, Wiley Online Library. (**c**) <i> Principle of tough interfacial bonding between hydrophobic triboelectric materials and hydrophilic hydrogels and <ii> the bonding properties, <iii> Schematic of the H-TENG and <iv> its performance [45]. Copyright 2020, The Royal Society of Chemistry. (**d**) <i> Schematic of hydrophobic ionic liquid Gel-Based TENG, <ii>, <iii> and <iv> performance of the prepared H-TENG in harsh environments [46]. Copyright 2020, American Chemical Society.

**Figure 3 polymers-14-01452-f003:**
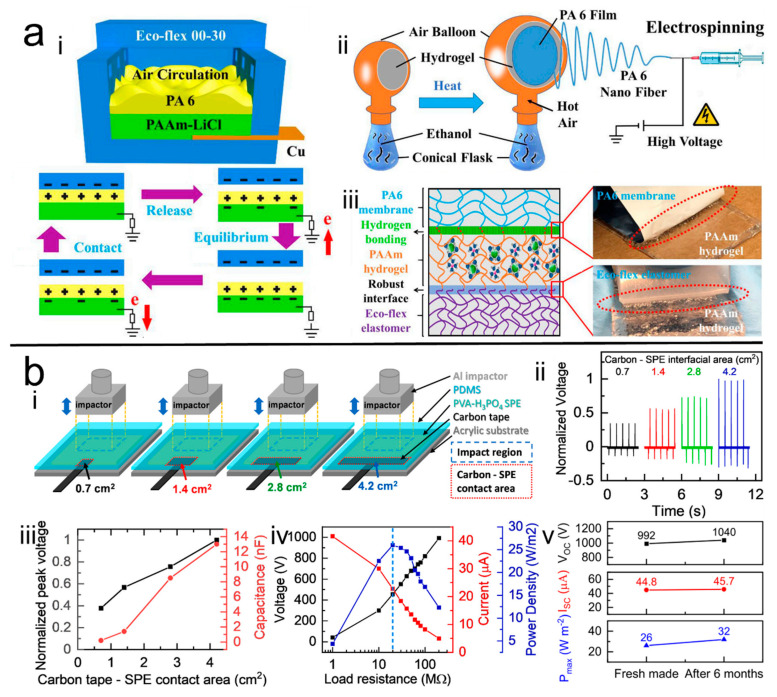
(**a**) <i> Schematic of the prepared H-TENG and <ii> the electrospinning process, <iii> the bonding surface of the prepared hydrogel [47]. Copyright 2020, Elsevier. (**b**) <i> Schematic of carbon tape and hydrogel contact area of the H-TENG, <ii>, <iii>, <iv> and <v> the performance of the prepared H-TENG [48]. Copyright 2021, Elsevier.

**Figure 4 polymers-14-01452-f004:**
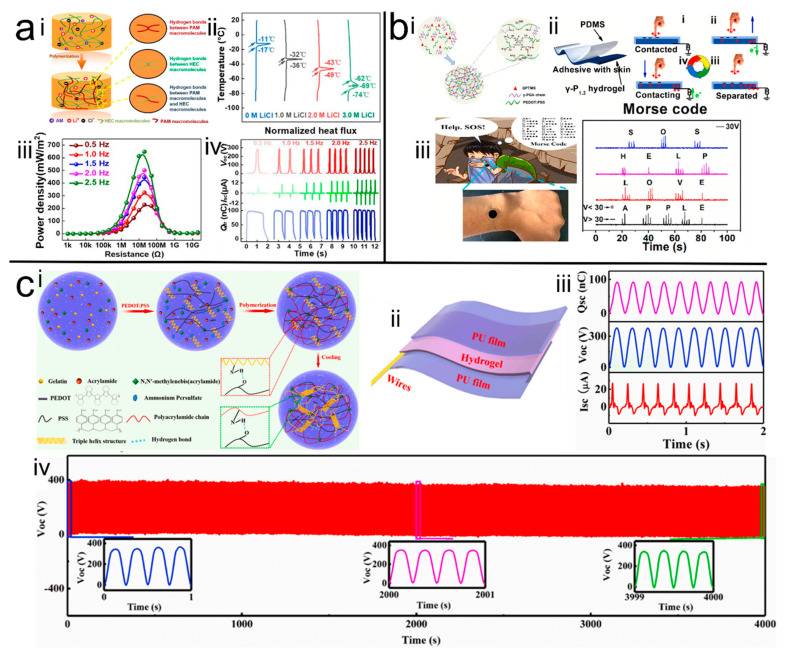
(**a**) <i> Schematic of the hydrogel, <ii> anti-freezing property of the prepared hydrogel, <iii> and <iv> the output performance of the H-TENG [52]. Copyright 2020, The Royal Society of Chemistry. (**b**) <i> and <ii> Schematic of the H-TENG, <iii> demonstration of Morse code communication [76]. Copyright 2022, Elsevier. (**c**) <i> and <ii> Schematic of the H-TENG, <iii> and <iv> performance of the prepared H-TENG [31]. Copyright 2020, Elsevier.

**Figure 5 polymers-14-01452-f005:**
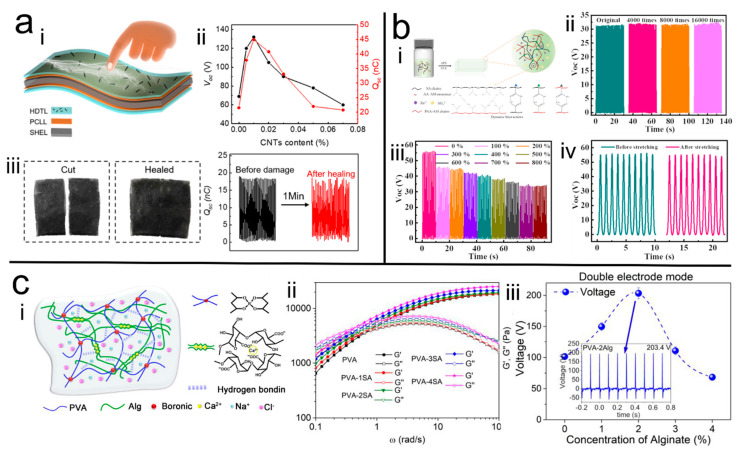
(**a**) <i> Schematic of the H-TENG, <ii> output performance related to the content of CNTs, <iii> self-healing property of the H-TENG [77]. Copyright 2020, Elsevier. (**b**) <i> Schematic of the hydrogel, <ii>, <iii> and <iv> the fatigue resistance and tensile performance of the H-TENG [54]. Copyright 2020, American Chemical Society. (**c**) <i> Schematic of the hydrogel, <ii> and <iii> the effect of hydrogel viscoelasticity on H-TENG performance [27]. Copyright 2020, American Chemical Society.

**Figure 6 polymers-14-01452-f006:**
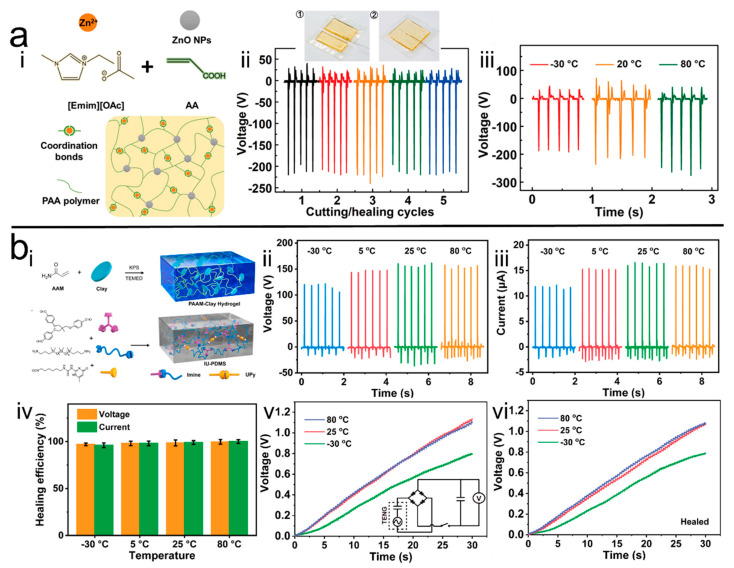
(**a**) <i> Schematic of the hydrogel, <ii> the self-healing property of H-TENG, <iii> environmental stability demonstration [57]. Copyright 2021, Elsevier. (**b**) <i> Schematic of the hydrogel, <ii>, <iii> the performance of H-TENG under harsh environments, <iv>, <v> and <vi> self-healing property of the H-TENG [59]. Copyright 2021, Elsevier.

**Figure 7 polymers-14-01452-f007:**
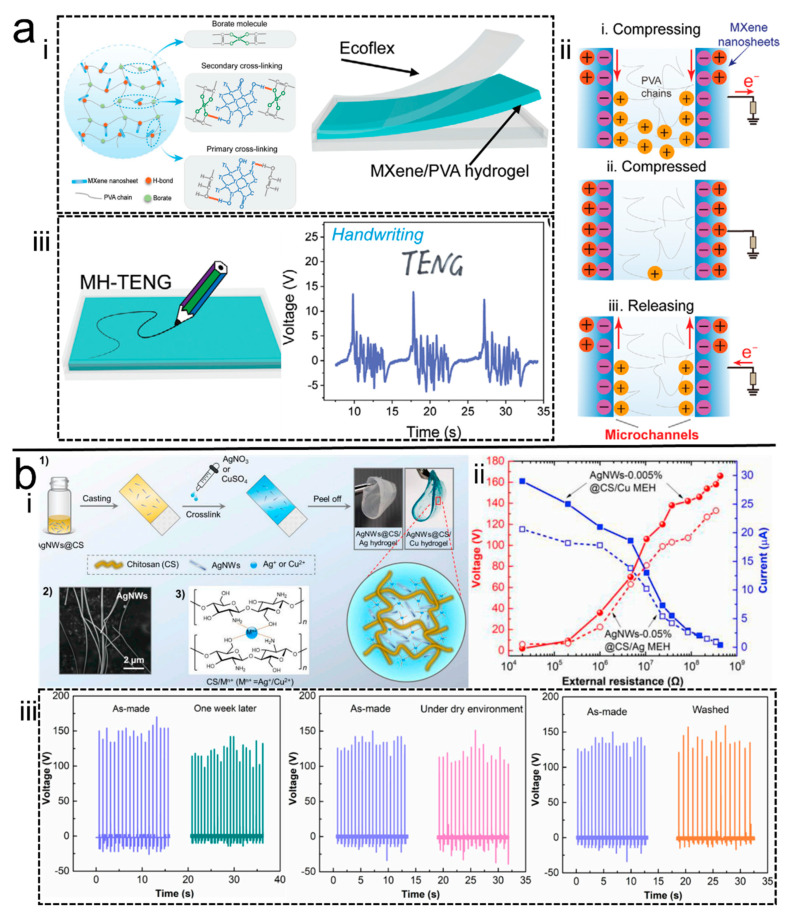
(**a**) <i> Schematic of the H-TENG, <ii> principle of the microchannels formed between MXene sheets, <iii> handwriting performance of the prepared H-TENG Flexible [60]. Copyright 2021, Wiley Online Library. (**b**) <i> Schematic of the hydrogel, <ii> output performance of the H-TENG, <iii> stability of the H-TENG [30]. Copyright 2019, Elsevier.

**Figure 8 polymers-14-01452-f008:**
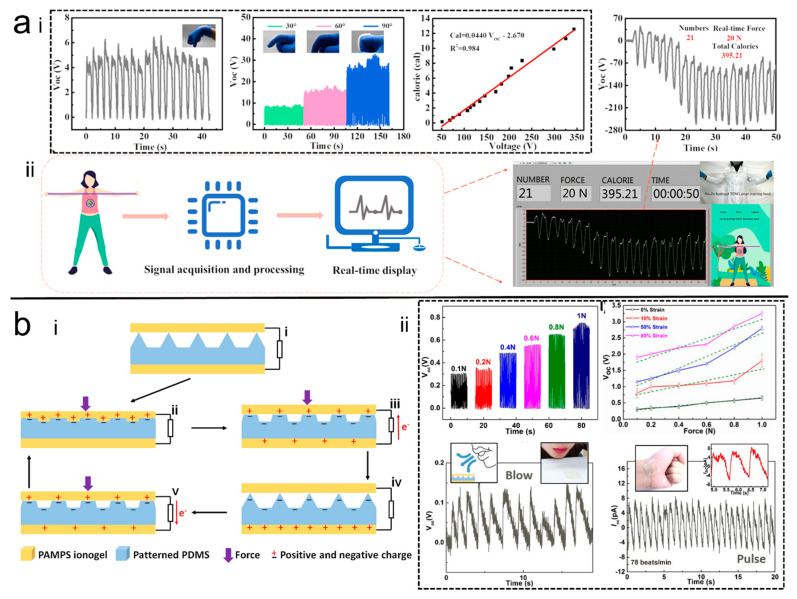
(**a**) <i> Sensing performance of the super stretchable H-TENG, <ii> Schematic of the Real-time display [54]. Copyright 2020, American Chemical Society. (**b**) <i> Schematic of the microstructure patterning H-TENG, <ii> sensing performance of the H-TENG [91]. Copyright 2019, Elsevier.

**Figure 9 polymers-14-01452-f009:**
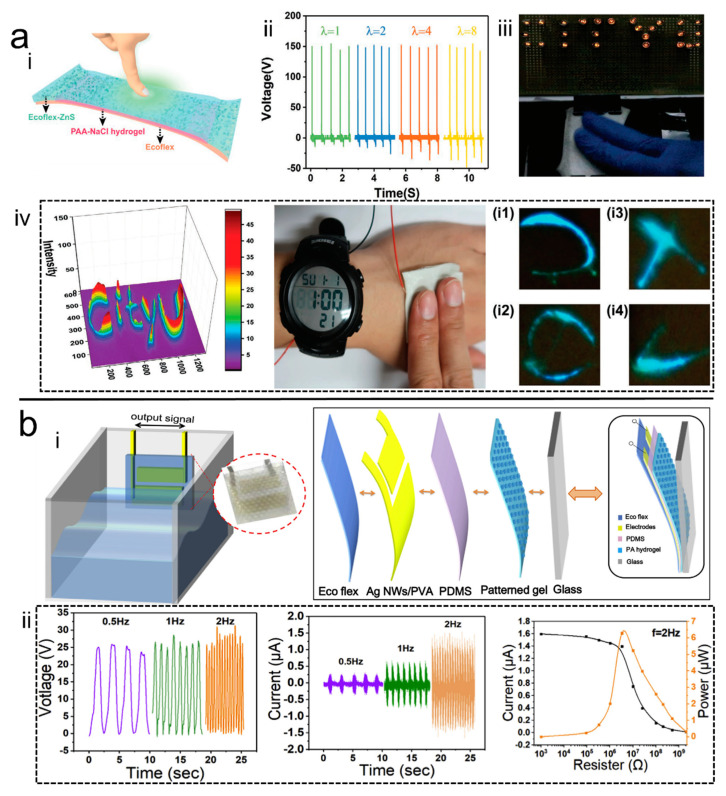
(**a**) <i> Schematic of the pressure visualization H-TENG, <ii> and <iii> output performance of the H-TENG, <iv> demonstration of the pressure visualization [51]. Copyrigth 2019, Wiley Online Library. (**b**) <i> Schematic of the H-TENG for water wave energy harvesting, <ii> the performance of the H-TENG [34]. Copyrigth 2019, Elsevier.

**Figure 10 polymers-14-01452-f010:**
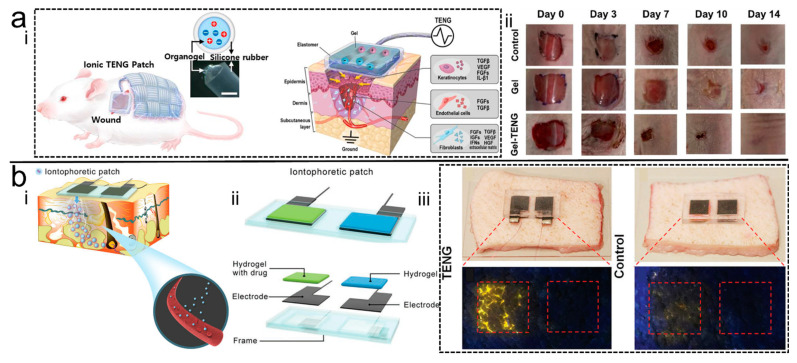
(**a**) <i> Schematic of H-TENG ionic patch, <ii> demonstration of accelerating wound healing process [88]. Copyright 2020, Elsevier. (**b**) <i> and <ii> Schematic of the drug delivery H-TENG, <iii> demonstration of the drug delivery property [93]. Copyright 2020, Wiley Online Library.

**Table 1 polymers-14-01452-t001:** Electrode classification and output performance of H-TENGs.

Hydrogel Material	Network Structure	Voc	Isc	Peak Power Density	Ref.
PVA	SN	200 V	22.5 µA	-	[43]
PAAm	SN	145 V	1.5 µA	35 mW/m^2^	[40]
PVA	SN	50 V	6.5 µA/cm^2^	40 µW/cm^2^	[44]
PAM	SN	311.5 V	32.4 µA	2.7 W/m^2^	[45]
PAM	SN	120 V	16.6 µA	0.51 mW/cm^2^	[46]
PAM	SN	170 V	11 µA	25 W/m^2^	[47]
PVA	SN	992 V	44.8 µA	26 W/m^2^	[48]
Cellulose	SN	187 V	0.51 µA	-	[49]
HA	SN	20 V	0.4 µA	5.6 mW/cm^2^	[50]
PAA	SN	180 V	65 µA	625 µW/cm^2^	[51]
PAM/HEC	MN	285 V	15.5 µA	626 mW/m^2^	[52]
PAM/PEDOT:PSS	MN	383.8 V	26.9 µA	1250 mW/m^2^	[31]
PAM/PDA	MN	230 V	12 µA	-	[53]
PVA/SA	MN	203.8 V	17.6 µA	0.98 W/m^2^	[27]
PAA-co-PAM/SA	MN	~55 V	~0.125 µA	-	[54]
PVA/HEC	MN	0.151 V	-	-	[38]
PAM/Alginate	MN	70 V	0.46 µA	135 mW/cm^2^	[55]
PAM/Cyclodextrin	MN	95 V	10 µA	0.64 mW/m^2^	[56]
PAA/ZnO	NC	216 V	41 µA	3.15 W/m^2^	[57]
PAM/Clay	NC	88.8 V	74.5 µA	75.4 mW/m^2^	[58]
PAM/Clay	NC	157 V	16 µA	710 mW/m^2^	[59]
PVA/MXene	NC	230 V	1.2 µA	0.33 W/m^2^	[60]
Chitosan/AgNWs	NC	218 V	34.4 mA/m^2^	2 W/m^2^	[30]
PVA/PDA/MWCNT	NC	95 V	1 µA	-	[61]
PAA/RGO/PANI	NC	21.8 V	48.7 µA	424 mW/m^2^	[62]
PAM/Silk Fibroin/GO/PEDOT:PSS	NC	12 V	0.4 µA	-	[63]
Bacterial Cellulose/ZnO	NC	57.6 V	5.78 µA	42 mW/m^2^	[64]
PAM/Graphene	NC	40 V	1.6 mA	0.3 W/cm^2^	[65]

Note: Voc represents open-circuit voltage and Isc is short-circuit current.

## Data Availability

The copyright permission of all figures was authorized.

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
