# Peer review of "Development and Applications of Hydrogel-Based Triboelectric Nanogenerators: A Mini-Review"

_polymers, 2022, doi:10.3390/polym14071452_

Round 1

Reviewer 1 Report

The authors presented a mini-review on the research and development of hydrogel-based triboelectric nanogenerators. The review is timely, concise and well written. It is easy to follow. The only point where I believe it could be improved is the concluding section 4, which is somewhat imprecise. It could be expanded and specified. In particular, it should be specified what kind of devices must be developed, what should be their characteristics (please specify numbers) and the necessary material properties quantified.

Author Response

Reply to reviewer’s comments for manuscript: (polymers-1661400) “Development and Applications of Hydrogel-Based Triboelectric Nanogenerators: A Mini-Review”.
We are deeply grateful for the comments and suggestions from the reviewers. They’ve helped us to improve the quality of our paper. Please see our detailed replies to the reviewers’ comments below.
Reviewer #1:
The authors presented a mini-review on the research and development of hydrogel-based triboelectric nanogenerators. The review is timely, concise and well written. It is easy to follow. The only point where I believe it could be improved is the concluding section 4, which is somewhat imprecise. It could be expanded and specified. In particular, it should be specified what kind of devices must be developed, what should be their characteristics (please specify numbers) and the necessary material properties quantified.

Reply to the comment:
We are grateful for the positive comments and the valuable suggestion on our paper from the reviewer. According to the reviewer’s comment, we have further revised the concluding section 4 as follows: “First, compared to the traditional electric power devices, recent reported H-TENGs displayed lower power density which could be further achieved via increasing its charge density by decreasing the dissipation of charges as well as enhancing the charge-trapping and charge-transportation ability of the conductive hydrogel by introducing the novel nanofillers. Second, the long-term stability of the H-TENGs should be carried out over months or even years before practical applications rather than only few hours in the laboratory. Finally, microstructured hydrogels are favorable for enhancing the outputting capacity of the H-TENGs via enhancing the contacting area; Moreover, it was also desirable to develop a feasible strategy to fabricate the H-TENGs on a scale way. Therefore, it is highly worthy of addressing to develop H-TENGs in the working principle as well as their material design, which also needs the joint efforts of researchers from different fields.”

Reviewer 2 Report

The submitted paper is a review on the triboelectric nanogenerators based on the hydrogel. It is divided into several sections that present strategies for fabrication of the hydrogel electrodes, their applications, and conclusions. In my opinion, the paper is valuable for researches working on the triboelectric nanogenerators. The following issues have to be improved or clarified before the paper can be accepted for publication:

  1. The recent achievements in the triboelectric nanogenerators should be summarized in the “Introduction” section. Following papers are recommended for Authors: Energy 238 C (2022) 122048, Nano Energy 89 (2021) 106316.
  2. It should be indicated the types of the power densities (peak or average values) are given in Table 1.
  3. Can be presented parameters other than open-circuit voltages and short-circuit currents? Authors are requested to consider to discuss various figures of merits of the triboelectric nanogenerators.
  4. Figures come from other publications. How about the copyrights and permissions for figures reproduction? Were they obtained by the Authors? If yes, the proper statements should be inserted into the figures captions.

Author Response

Reply to reviewer’s comments for manuscript: (polymers-1661400) “Development and Applications of Hydrogel-Based Triboelectric Nanogenerators: A Mini-Review”

We are deeply grateful for the comments and suggestions from the reviewers. They’ve helped us to improve the quality of our paper. Please see our detailed replies to the reviewers’ comments below.

Reviewer #2:

The submitted paper is a review on the triboelectric nanogenerators based on the hydrogel. It is divided into several sections that present strategies for fabrication of the hydrogel electrodes, their applications, and conclusions. In my opinion, the paper is valuable for researches working on the triboelectric nanogenerators. The following issues have to be improved or clarified before the paper can be accepted for publication.

  1. The recent achievements in the triboelectric nanogenerators should be summarized in the “Introduction” section. Following papers are recommended for Authors: Energy 238 C (2022) 122048, Nano Energy 89 (2021) 106316.

Reply to the comment:

We appreciate the reviewer for the valuable suggestion. Suggested references have been cited and relative modifications have been given in our revised manuscript as follows: “TENGs can harvest mechanical energy that is ubiquitous in human daily life including wave, wind, as well as human motion such as respiration and body movements, which have unique superiority in simple design, low costs, feasible portability over other power sources [10,11].”

  1. It should be indicated the types of the power densities (peak or average values) are given in Table 1.

Reply to the comment:

We are sorry to make the confusion. The power densities in table 1 were the peak density. Therefore, in table 1 in the revised manuscript, the power density has been replaced by “Peak power density”.

  1. Can be presented parameters other than open-circuit voltages and short-circuit currents? Authors are requested to consider discussing various figures of merits of the triboelectric nanogenerators.

Reply to the comment:

Thank for the suggestion. Besides the open-circuit voltages and short-circuit currents, other parameters including mechanical performance, viscoelasticity, self-healing properties as well as transmittance of the hydrogel were also discussed on the performance of the H-TENGs as follows:

In section 2.1, the sentence “This TENG demonstrated 1160 % stretchability and 96.2 % transparency.” was used to display the stretchability and transparency of H-TENG. Moreover, the sentence “which enabled the TENG to self-heal and to obtain an effective and stable output signal even after self-healing; meanwhile, the TENG also displayed 700 % stretchability and long-term durability for 1000 continuous cycles of voltage output measured at 15-day intervals.” was used to demonstrate the self-healing property, mechanical performance and stability.

In section 2.2, the sentence “It was proved that the self-healable hydrogel endows the healing ability of the H-TENGs, whose output performance could reach 94% after healing.” was used to indicate the self-healing property of H-TENG. Moreover, the sentence “which was synthesized by one-step radical polymerization of acrylamide monomer in aqueous hydroxyethyl cellulose solution, which can tolerate -69 °C without freezing after the addition of lithium chloride.” was used to indicate the anti-freezing property of H-TENG.

In section 2.3, the sentence “The H-TENG prepared using this NC hydrogel as electrode demonstrated an ultra-wide temperature operating range from -30 °C to 80 °C and a long-term stability of more than 30 days.” was used to prove the H-TENG can operate in harsh environments.

More relative discussions are highlighted in red in the revised manuscript.

In addition, we have modified Figure 6a to avoid displaying the open-circuit voltage and short-circuit current in too many figures.

  1. Figures come from other publications. How about the copyrights and permissions for figures reproduction? Were they obtained by the Authors? If yes, the proper statements should be inserted into the figure captions.

Reply to the comment:

Thank you for the suggestion. The copyrights and permissions for the figures reproduction of the figures have been permitted by the publishers. Relative modifications have been given in the image captions in the revised manuscript.
